# Overwintering Camelina and Canola/Rapeseed Show Promise for Improving Integrated Weed Management Approaches in the Upper Midwestern U.S.

**DOI:** 10.3390/plants12061329

**Published:** 2023-03-15

**Authors:** Wun S. Chao, James V. Anderson, Xuehui Li, Russ W. Gesch, Marisol T. Berti, David P. Horvath

**Affiliations:** 1USDA-ARS, Edward T. Schafer Agricultural Research Center, Fargo, ND 58102, USA; james.v.anderson@usda.gov (J.V.A.); david.horvath@usda.gov (D.P.H.); 2Plant Sciences Department, North Dakota State University, Fargo, ND 58108, USA; xuehui.li@ndsu.edu (X.L.); marisol.berti@ndsu.edu (M.T.B.); 3USDA-ARS, North Central Soil Conservation Research Laboratory, Morris, MN 56267, USA; russ.gesch@usda.gov

**Keywords:** camelina (*Camelina sativa* L.), canola/rapeseed (*Brassica napus* L.), freezing tolerance, genotyping, oilseed crops, overwintering, weed suppression

## Abstract

Winter oilseed cash cover crops are gaining popularity in integrated weed management programs for suppressing weeds. A study was conducted at two field sites (Fargo, North Dakota, and Morris, Minnesota) to determine the freezing tolerance and weed-suppressing traits of winter canola/rapeseed (*Brassica napus* L.) and winter camelina [*Camelina sativa* (L.) Crantz] in the Upper Midwestern USA. The top 10 freezing tolerant accessions from a phenotyped population of winter canola/rapeseed were bulked and planted at both locations along with winter camelina (cv. Joelle) as a check. To phenotype our entire winter *B. napus* population (621 accessions) for freezing tolerance, seeds were also bulked and planted at both locations. All *B. napus* and camelina were no-till seeded at Fargo and Morris at two planting dates, late August (PD1) and mid-September (PD2) 2019. Data for winter survival of oilseed crops (plants m^−2^) and their corresponding weed suppression (plants m^−2^ and dry matter m^−2^) were collected on two sampling dates (SD) in May and June 2020. Crop and SD were significant (*p* < 0.05) for crop plant density at both locations, and PD in Fargo and crop x PD interaction in Morris were significant for weed dry matter. At Morris and Fargo, PD1 produced greater winter *B. napus* survival (28% and 5%, respectively) and PD2 produced higher camelina survival (79% and 72%, respectively). Based on coefficient of determination (*r*^2^), ~50% of weed density was explained by camelina density, whereas ≤20% was explained by *B. napus* density at both locations. Camelina from PD2 suppressed weed dry matter by >90% of fallow at both locations, whereas weed dry matter in *B. napus* was not significantly different from fallow at either PD. Genotyping of overwintering canola/rapeseed under field conditions identified nine accessions that survived at both locations, which also had excellent freezing tolerance under controlled conditions. These accessions are good candidates for improving freezing tolerance in commercial canola cultivars.

## 1. Introduction

The chemical, mechanical, and cultural measures used to control weeds require land managers and taxpayers to expend hundreds of millions of U.S. dollars annually [1,2,3]. Integrated weed management (IWM), which utilizes a combination of control tactics, is considered the most effective approach to managing weeds. Incorporation of cover crops into integrated weed management systems is gaining popularity for reducing the spread of weeds and evolution of herbicide resistance resulting from conventional monocropping practices. As an example, cover crop acreage in the U.S. increased by an average of 50% from 2012–2017, which represented 6.23 million hectares in 2017 [4]. In North Dakota alone, cover crop acreage increased by 89% (from 86,526 to 163,601 hectares) during this same 5-year period. This increase also resulted from the value-added ecosystem benefits provided by cover crops, which includes the suppression of weeds, improved pollinator habitat and soil health, reduced soil erosion and nutrient losses, and climate change mitigation and adaptation [5,6,7].

Some winter oilseed species have been identified as regionally appropriate winter cash crops or cover crops. However, in the Northern Great Plains (NGP) and Upper Midwest (UMW) regions of the U.S., the number of economically viable oilseed crops or cover crops that consistently tolerate the harsh winter conditions experienced in these regions is limited. Camelina [(*Camelina sativa* (L.) Crantz] is a member of the Brassicaceae family that has shown promise as a winter-hardy cash or cover crop; winter biotypes of camelina have excellent freezing tolerance traits and can survive the winter conditions experienced in the NGP [8,9,10]. Camelina has also been reported to have a relatively short period from seeding to harvest [8], with seed yields ranging from 743 to 2300 kg ha^−1^ across Minnesota and North Dakota [11,12,13]. Based on these traits, winter camelina is being considered as a viable oilseed cash cover crop for developing multi-cropping systems [14].

Because canola (*Brassica napus* L.) has better yield and established markets compared with camelina, identifying more freezing tolerant winter canola/rapeseed could be an excellent option for developing new rotational cropping systems and IWM approaches in the NGP and UMW. Winter canola generally has 20–30% greater yield potential than spring canola [15]; winter canola seed yield typically exceeds 2000 kg ha^−1^ and can reach 5000 kg ha^−1^ [16]. In addition, it provides spring-time pest suppression [17,18] and creates the opportunity to expand winter canola acreage on millions of hectares that are not planted due to extreme weather conditions. Identification of germplasm better adapted to winter conditions could also reduce winterkill in traditional areas of winter canola production like Idaho, Oklahoma, and Kansas.

The objective of this study was to determine the freezing tolerance and weed-suppressing traits of winter canola/rapeseed and winter camelina in the Upper Midwestern USA. Field trials indicated that a winter biotype of camelina (cv. Joelle) and some canola/rapeseed accessions can survive the harsh winter conditions in this region, and increased survival of winter oilseed crops to extended freezing conditions have a direct correlation on the suppression of weeds.

## 2. Results and Discussion

### 2.1. Significant Factors

Crop (C) and Sampling Date (SD) were significant (*p* ≤ 0.05) for crop plant counts at both locations, while Planting Date (PD), and C × PD, and C × SD interactions were significant factors at the Fargo but not the Morris field site (Table 1). For weed counts and dry matter biomass, only PD was significant for biomass in Fargo, while the Crop × PD interaction for dry matter was significant in Morris (Table 2).

### 2.2. Freezing Tolerance of Canola/Rapeseed and Camelina under Field Conditions

To examine the overall freezing tolerance of the canola/rapeseed population, both the top 10 freezing tolerant rated winter canola/rapeseed accessions previously identified and the 621 accessions from Kansas State University (KSU) and the Leibniz Institute of Plant Genetics and Crop Plant Research (IPK) [19,20] were evaluated under field conditions at both Fargo and Morris. As a check, freezing tolerant winter camelina ‘Joelle’ was also planted at each location for comparison. Plant densities for camelina and canola/rapeseed accessions demonstrated varying degrees of overwinter survival at both locations (Figure 1 and Appendix A). Although reduced plant densities were observed in the spring of 2020 as compared with the fall 2019, we did observe greater winter survival of canola/rapeseed from planting date 1 (PD1-late August) at both locations (Figure 1A,B). At the Fargo location, a total of 75 canola/rapeseed plants overwintered; however, many more canola/rapeseed plants overwintered at Morris. These results indicate that earlier fall plantings improved winter survival under field conditions, which may be associated with increased growth and development. Regardless, this data demonstrates that germplasm from selected winter varieties of canola/rapeseed have potential to survive the harsh winter condition of the NGP and the UMW. In contrast, many more winter camelina (cv. Joelle) plants survived from the second planting date as opposed to the first date (Figure 1A,B). These results are consistent with the study done by Wittenberg et al. [10] who reported that winter camelina sown before September usually results in poor winter survival in the UMW.

Results clearly revealed that planting date of winter oilseed crops affects winter survival in ND and MN, and the optimal planting dates appear to be different among different species. The better winter survival for winter canola/rapeseed at Morris vs. Fargo could be due to different planting systems (no-till wheat stubble vs. no-till bare soil). Although the daily minimum winter temperatures appear similar between Fargo and Morris (Figure 2), the total freezing degrees below −10 °C in Fargo (−1794) vs. Morris (−1556) from 23 October 2019 to 4 April 2020 (Figure 2 and Appendix A) indicate that Fargo experienced 238 more freezing degrees than Morris. Regardless of the greater freezing degrees for air temperature in Fargo, soil temperature in Morris was colder than that of Fargo from December 2019 to April 2020 (Figure 2), which may be due to greater snow cover on the ground in Fargo vs. Morris from December to April (see Daily Snow Depth in Appendix A). Because more winter canola/rapeseed survived at Morris vs. Fargo (Figure 1), our data indicate that lower air temperature but not soil temperature (Figure 2) was the main factor explaining winter survival.

### 2.3. Weed Suppression by Overwintering Canola/Rapeseed and Camelina

Winter camelina (cv. Joelle) consistently survives the harsh winter conditions of the UMW [10], whereas canola/rapeseed germplasm generally does not show freezing tolerance. In North Dakota, spring planted canola has been reported to suppress up to 60–90% of mid− to late−season weeds compared to fallow [13]. Components contributing to the greater weed competitiveness of canola cultivars have been studied, and tactics that can be employed to suppress weeds include heterosis, higher seeding rates, optimal row spacing, early−season crop biomass accumulation, plant height, and allelopathy [21,22,23,24,25]. Thus, identifying freezing tolerant varieties of winter canola/rapeseed could help meet the goal of both expanding U.S. acreage while suppressing early season weed establishment and potentially the spread of herbicide resistant weeds.

In this study, we examined weed suppression capacity based on overwintering winter canola/rapeseed and camelina in Fargo and Morris. Results demonstrated that overwinter survival of winter camelina (plant density) correlates well with increased spring weed suppression (Table 3 and Appendix A); fall-planted winter camelina produced greater plant counts the following spring compared with plant counts observed for winter canola/rapeseed at both field sites. The stronger correlation between plant counts of weeds and winter camelina in Morris and Fargo (*r* = −0.67 and −0.70, respectively) compared with plant counts of weeds and winter canola/rapeseed (r = −0.66 and −0.45, respectively) appears to result from a greater winter survival of winter camelina. Moreover, the coefficient of determination (*r*^2^) values (Table 3) suggests that stand density for camelina and weeds in Morris and Fargo better fitted to the regression model than that of the stand density for canola/rapeseed and weeds. Specifically, approximately 50% of the variation in weed counts was explained by camelina density (*r*^2^ = 0.45 and 0.50 in Morris and Fargo, respectively), whereas only about 20% of variation in weed counts was explained by canola/rapeseed density (*r*^2^ = 0.18 and 0.20 in Morris and Fargo, respectively). Except for winter canola/rapeseed plant density vs. weed DM (*r*^2^) at Fargo, similar results were also obtained comparing plant densities of winter canola/rapeseed and camelina with DM biomass of weeds in both Morris and Fargo (Table 3, and Appendix A). In summary, field trials support the hypothesis that increased survival of winter camelina or canola/rapeseed to extended freezing conditions could have a direct correlation on suppression of weeds. The suppression of weed DM (g m^−2^) observed in the camelina plots at both locations was greater than 90% of the fallow plots for PD2 (Figure 3, and Appendix A).

### 2.4. Genotyping of Winter Canola/Rapeseed That Survived Winter under Field Conditions

Freezing tolerance is a challenging trait to study in natural conditions due to unpredictability of the weather conditions; therefore, several genome-wide association studies for freezing tolerance in canola/rapeseed were done in controlled conditions [19,20,26,27]. However, field study is essential to determine if those freezing tolerance accessions identified in controlled conditions would still function analogously under natural conditions. In this study, mixed canola/rapeseed accessions were planted at two field sites (Fargo and Morris), and the genetic constitutions of these accessions that survived the winter conditions were examined by GBS to determine their genotype. A total of 75 overwintered plants from Fargo and 75 overwintered plants from Morris were genotyped and compared with the genetic constitution of 621 accessions (KSU and IPK populations) based on the genetic pair-wise distance using 1587 SNP markers (Appendix A). The genetic constitution of overwintered individual plants matches with 58 canola/rapeseed accessions based on their genetic pair-wise distance values of less than 0.1. Among these 58 accessions, only nine overwintered in both Fargo and Morris (Table 4 and Appendix A). These nine accessions are rather similar in genetic constitution based on a neighbor-joining tree (Appendix A) and a distance matrix (genetic pair-wise distance <0.2, see Appendix A) analysis, and the genetic constitutions of ARS165, ARS228, and ARS229 are extremely similar (Appendix A). Notably, all nine of these accessions were also categorized to be freezing tolerant according to our previous freezing chamber studies [19] (visual damage scale and chlorophyll fluorescence Fv/Fo are provided in Table 4).

The nine accessions that survived at both locations had 716 loci with significantly different allelic frequencies as compared to the entire *B. napus* population. Forty-eight groups of 5 or more closely linked loci clustered across 12 chromosomes with Chromosomes A01 containing 9 clusters, followed by A05 with 7, A03 and A06 with 6 each, C06 with 4 and C03 with 3 clusters (Appendix A). These clusters may contain genes that are responsible for overwintering. Interestingly, some SNPs (SA06_2471189, SA06_2477656, SC06_1107946, and SC06_1138598) within the clusters on chromosome A06 and C06 (highlighted in green in Appendix A, under the column “position”) are direct matches to SNPs identified from previous association studies during acclimation and deacclimation in a *B. napus* population [26]. Another SNP (SA03_16201884) situated outside of these clusters on chromosome A03 (highlighted in blue in Appendix A, under the column “position”) is also a direct match to a SNP associated with freezing tolerance in a European *B. napus* population [20], and a few gene models such as *9-CIS-EPOXYCAROTENOID DIOXYGENASE 3*, *CTC-INTERACTING DOMAIN 9*, *TERPENE SYNTHASE 20*, and several *CYTOCHROME P450* are located around this SNP marker.

Three accessions, ARS012, ARS029, and ARS246, had three or more overwintered canola/rapeseed individuals with matched genotype (genetic pair-wise distance values less than 0.1) in both Fargo and Morris, and they thus should be exceptionally freezing tolerant. Since the ability of winter oilseed crops to provide maximum ecosystem services depends on their winter survival, ARS012, ARS029, and ARS246 are thus good candidates for breeding freezing tolerant commercial canola cultivars.

## 3. Materials and Methods

### 3.1. Plant Materials

A population of 621 accessions consisting of mostly winter canola/rapeseed was obtained from KSU (399 accessions) and Leibniz Institute of Plant Genetics and Crop Plant Research (IPK) in Gatersleben, Germany (222 accessions). The original names for the KSU and IPK accessions are available in Appendix A along with our designated internal code names for each accession (i.e., ARSXXX). The KSU and IPK populations were previously phenotyped for freezing survival, and the freezing tolerance capacity of each accession was rated based on a visual damage scale and chlorophyll fluorescence (Fv/Fo and Fm/Fo) under controlled conditions [19,20]. For weed suppression and freezing survival studies in the field, seeds from the top ten freezing tolerant accessions from the KSU population [19] were bulked for fall planting at the North Dakota State University main campus field site, Fargo, ND, USA (46.89790776592843° N, −96.81765200851491° W) and at the Swan Lake Research Farm, Morris, MN (45°41′06.2″ N 95°47′54.0″ W). Soils at Fargo consist of the Fargo soil series (fine, smectitic, frigid Typic Epiaquerts), and soils at Swan Lake consist of the Barnes loam soil (fine-loamy, mixed, superactive, frigid Calcic Hapludoll). The 10 accessions included ARS012, ARS029, ARS32, ARS164, ARS189, ARS191, ARS209, ARS229, ARS246, and ARS396 (Appendix A). To determine the overall freezing survival among our total winter canola/rapeseed population (621 accessions), seeds from each accession were also bulked and fall planted in Fargo, ND and Morris, MN, USA. Seed of camelina winter biotype ‘Joelle’ was fall planted at the Fargo, ND and Morris, MN field sites as a check. As mentioned in the introduction, Joelle winter camelina is known to be very freeze hardy.

### 3.2. Freezing Tolerance vs. Weed Suppression

Seeds of the top 10 winter-hardy canola/rapeseed accessions and Joelle winter camelina were planted in a randomized complete block design with three replicates. Individual plot size was 6.1 m × 2 m. Each canola/rapeseed plot was seeded with 5.8 g of bulked winter canola/rapeseed consisting of 0.58 g per accession, to give a field planting rate of 5.0 kg ha^−1^ using 0.30 m row spacing. Each location had a total of six winter canola/rapeseed plots with three replicated plots planted in late August 2019 (PD1; 30 and 26 August, respectively, in Fargo and Morris) and three replicated plots planted in mid-September 2019 (PD2; 17 and 24 September, respectively, in Fargo and Morris). Winter camelina (cv. Joelle) was also planted on PD1 and PD2 at each location, with three replicates. Camelina was planted at a rate of 6.7 kg ha^−1^ using 0.15 m row spacing, which is generally a recommended rate and row spacing for its production. All seeds were no-till drill planted into wheat (*Triticum aestivum* L.) stubble at Morris or into no-till soil directly at Fargo. Each location also included three replicated control plots (fallow) per planting date (six total fallow plots) with no canola/rapeseed or camelina. All plots seeded with canola/rapeseed were broadcast fertilized with 22.4–33.6 kg ha^−1^ N in the fall of 2019 and 78.5 kg ha^−1^ in spring 2020 after plant growth had resumed. Camelina plots were broadcast with 67.3 kg ha^−1^ N in spring 2020 after plants resumed growth. Fallow control plots received no fertilizer. Canola/rapeseed and camelina plant counts (plant m^−2^) were taken on 1 m of the center rows of plots in the fall of 2019 (6 November and 31 October, respectively) in Fargo and Morris. The 1 m of row used for plant counts was marked with flags, so that counts for the following spring could be made from the same area. Overwinter survival of canola/rapeseed and camelina was recorded 18 and 15 May 2020, and 29 and 30 June 2020 in Fargo and Morris, respectively. Weed plant counts from two randomly placed 0.25 m^2^ quadrants plot^−1^ were taken on the same May dates listed above and the weed dry matter biomass was obtained from two 0.25 m^2^ quadrants plot^−1^ on the June 2020 dates listed above.

### 3.3. Freezing Survival Evaluation of Winter Rapeseed Population

To evaluate the freezing tolerance of our winter rapeseed population (KSU + IPK populations; 621 accessions) under field conditions, four bulked samples (5 seeds from each accession or approximately 2500 seeds/bulked sample) were planted into an 18.3 m × 2 m plot at both locations on the two 2019 autumn planting dates described above. The seeding rate for the bulked canola/rapeseed population was 3.7 kg ha^−1^ using 6 rows at 0.30 m row spacing in no-till wheat stubble at Morris and no-till soil at Fargo. Nitrogen fertilizer was applied the same as that mentioned above. In the spring of 2020, leaf tissue samples of overwintered canola/rapeseed accessions were collected prior to bolting and flash frozen in liquid nitrogen and then stored at −80 °C for genotyping. There was a total of 75 canola/rapeseed plants that overwintered in Fargo, and all were genotyped. However, many more canola/rapeseed plants overwintered at Morris; thus, only 75 plants were randomly chosen for genotyping.

### 3.4. Environmental Data for Field Sites

Temperature and precipitation data (2019–2020) for the Fargo and Morris field sites, respectively, were obtained from the North Dakota Agricultural Weather Network https://ndawn.ndsu.nodak.edu, accessed on 31 January 2023, and the Swan Lake Research Farm field site https://www.ars.usda.gov/midwest-area/morris-mn/soil-management-research/docs/weather/, accessed on 31 January 2023.

### 3.5. Genotyping of Accessions for Identification of Overwintering Canola/Rapeseed Plants

Canola/rapeseed plants that survived the wintering conditions at both locations were genotyped using genotyping-by-sequencing (GBS). Leaf tissue (~50 mg) was lysed in 2 mL tubes with Lysing Matrix Z utilizing a FastPrep-96 Homogenizer with CoolPrep™ adapter (MP Biomedicals, Irvine, CA, USA, Prod #s 116961050-CF, 116010500, & 116002528, respectively) and dry ice at 1800 rpm for 30 s. The DNA was extracted with DNeasy Plant Mini Kit (Qiagen, Germantown, MD, USA, Prod # 69104) and eluted in 200 µL elution buffer and then quantified with a Quant-iT PicoGreen dsDNA assay kit (P7589; Thermo Fisher Scientific, Waltham, MA, USA). The GBS library was constructed using the same protocol described in Horvath et al. [28]. The GBS library was sequenced on an Illumina (San Diego, CA, USA) HiSeq 4000 to generate single-end, 100-bp reads at the Genomic Sequencing and Analysis Facility, University of Texas Southwestern Medical Center, Dallas, TX. The SNP discovery and genotype calling was performed with the TASSEL-GBS pipeline [29] using the *Brassica napus* v4.1 [30] as a reference genome. A total of 1587 SNP markers with missing values of less than 10% and minor allele frequency greater than 5% were obtained. For accession identification, pair-wise genetic distance of overwintered plants and 621 accessions originally obtained from the KSU and IPK *B. napus* populations [19,20] were calculated as “1-IBS (identical by state) similarity” using the 1587 GBS SNP markers with TASSEL 5.0 [31]. A distance value of less than 0.1 indicates that the two individuals have a high probability of being from the same lineage. Neighbor-joining tree and distance matrix for 58 overwintered winter canola/rapeseed accessions were performed with TASSEL 5.0 [31] using all 69,554 SNP markers with default parameters. Over-represented alleles among the top 9 genotypes were determined by chi square analysis using the genotyping data from all 621 genotypes in our collection (Appendix A). Briefly, the number of major, minor (including minor alleles marked as deletions), and missing alleles were quantified in the entire population to generate the expected ratios of those three classes of alleles at each of the 69,554 loci. Likewise, numbers of each of those allelic groups were quantified in the 9 genotypes with superior winter survival. Chi square statistics were generated for each locus with no missing alleles and a chi square statistic less than 0.01 (with 2 degrees of freedom).

### 3.6. Statistical Analysis

The Mixed Procedure of SAS was used to analyze data collected from this study, which included three replicates (rep), two planting dates (PD), and three sampling dates (SD) at two locations. Data collected for evaluation included crop plant counts and weed plant counts and dry matter biomass. For the ANOVA, location, crop, PD, and SD were fixed and independent variables and plant count and biomass were dependent variables. When ANOVA showed significant differences (*p* ≤ 0.05), an *F*-protected LSD at α = 0.05 was used to differentiate treatment means. Locations were analyzed separately.

## 4. Conclusions

Although winter hardy oilseed cash cover crops are gaining popularity as components of integrated weed management systems, it is challenging to obtain economically viable oilseed/cover crops that consistently tolerate the harsh winter conditions in the NGP and UMW. In this research, winter-hardy *B. napus* accessions were examined in Fargo, ND and Morris, MN along with a winter biotype of camelina (cv. Joelle) as a check to determine their fitness on overwinter survival and weed suppression. Field trials indicated that some *B. napus* accessions can survive the harsh winter conditions in this region, and increased survival of *B. napus* to extended freezing conditions have a direct correlation on suppression of weeds. Thus, it is imperative to use freezing tolerant *B. napus* germplasm that can survive the winter conditions of the NGP and UMW for weed suppression. However, even if some freezing tolerant *B. napus* germplasm can overwinter in this area, there is still room for improvement such as selecting optimal planting dates and applying appropriate planting systems such as no-till wheat stubble.

## Figures and Tables

**Figure 1 plants-12-01329-f001:**
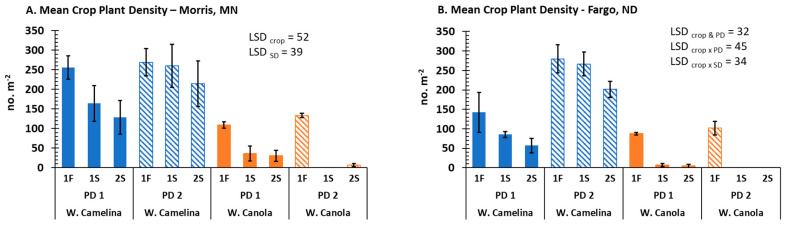
Plant densities of winter camelina (cv. Joelle) and 10 bulked accessions of the most freezing tolerant winter canola/rapeseed (W. Canola) at Morris, MN (**A**) and Fargo, ND (**B**). Plant densities were recorded prior to winter (1F) and in spring of 2020 (1S, 2S). Error bars represent the standard error of mean.

**Figure 2 plants-12-01329-f002:**
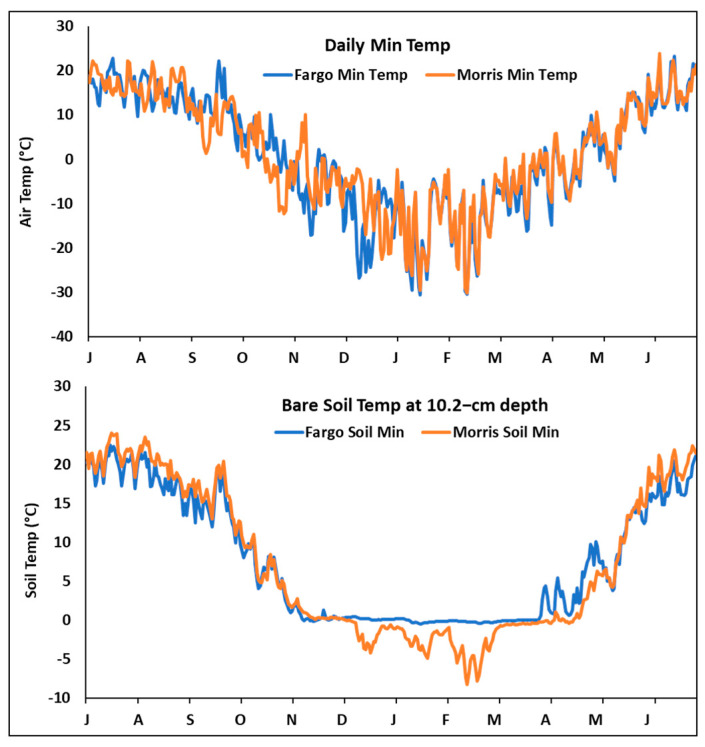
Comparison of minimum average daily air and soil temperatures (at 10.2−cm depth) from July 2019 through June 2020 in Fargo, ND (https://ndawn.ndsu.nodak.edu, accessed on 31 January 2023) and Morris, MN (https://www.ars.usda.gov/midwest-area/morris-mn/soil-management-research/docs/weather/, accessed on 31 January 2023).

**Figure 3 plants-12-01329-f003:**
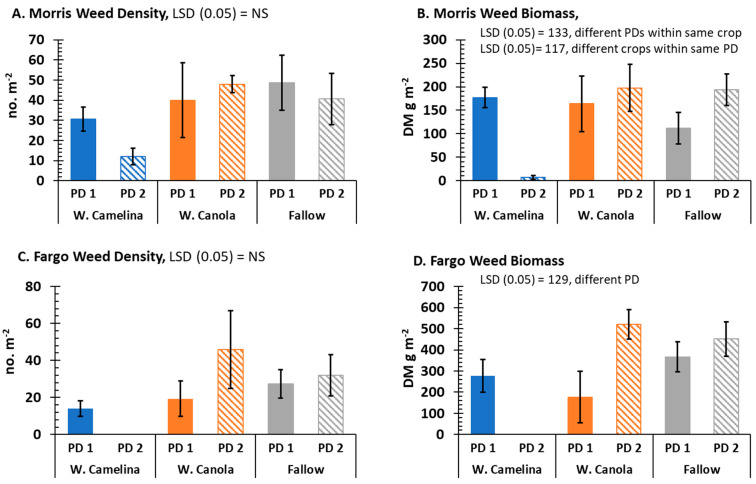
Weed density in crop [winter camelina, winter canola/rapeseed (W. Canola)] vs. fallow weed density (**A**,**C**) or weed dry matter (DM) biomass in crop [winter camelina, winter canola/rapeseed (W. Canola)] vs. fallow weed DM biomass (**B**,**D**) at Morris, MN (**A**,**B**) and Fargo, ND (**C**,**D**) based on crop planting date 1 (PD1 = late August 2019) or PD2 (mid−September 2019). Bars represent the mean ± S.E.

**Table 1 plants-12-01329-t001:** ANOVA table showing level of significance (*p*-values) for crop counts (no. m^−2^) at Fargo, ND and Morris, MN field sites.

	Fargo, ND	Morris, MN
SOV	Crop (no. m^−2^)	
Crop (C)	0.0003	0.0010
Planting Date (PD)	0.0056	NS
C × PD	0.0059	NS
Sampling Date (SD)	<0.0001	<0.0001
C × SD	0.0488	NS
PD × SD	NS	NS
C × PD × SD	NS	NS

**Table 2 plants-12-01329-t002:** ANOVA table showing level of significance (*p*-values) for weed counts (no. m^−2^) and dry matter (DM g m^−2^) at Fargo, ND and Morris, MN field sites.

	Fargo, ND		Morris, MN	
SOV	Weed (no. m^−2^)	Weed DM (g m^−2^)	Weed (no. m^−2^)	Weed DM (g m^−2^)
Crop (C)	NS	NS	NS	NS
Planting Date (PD)	NS	0.0266	NS	NS
C × PD	NS	NS	NS	0.0163

**Table 3 plants-12-01329-t003:** Regression analysis and correlation between crop plant density (#) and weed density (#) and dry matter biomass (DM) at Fargo, ND and Morris, MN.

Correlation		*r*	*r* ^2^	*r*	*r* ^2^
Crop	Weed	Fargo, ND		Morris, MN	
Camelina #	#	−0.70	0.50	−0.67	0.45
	DM	−0.70	0.50	−0.65	0.42
*B. napus* #	#	−0.45	0.20	−0.66	0.18
	DM	−0.75	0.57	−0.41	0.17

**Table 4 plants-12-01329-t004:** Summary of genotypic determination of winter canola/rapeseed plants that survived the 2019–2020 winter in Fargo, ND (F) and Morris, MN (M). A distance matrix for the overwintered canola/rapeseed individuals (Appendix A) was used to identify genotypes. The values for visual damage scale and chlorophyll fluorescence (Fv/Fo) were determined previously [19,20]. Accessions in black color were from the top 10 freezing tolerant accessions from a KSU population [19], and those in blue color represent other accessions among the total winter rapeseed population (621 accessions).

Accession	ARS012	ARS029	ARS165	ARS191	ARS209	ARS228	ARS229	ARS246	ARS325
survived	F = 7	F = 5	F = 6	F = 2	F = 2	F = 6	F = 6	F = 12	F = 7
	M = 10	M = 3	M = 1	M = 5	M = 2	M = 1	M = 1	M = 7	M = 2
Visual damage	1.9	2.4	1.9	2.1	2.1	1.7	2.4	2.0	2.2
Fv/Fo, 3 day	3.9	3.6	4.0	4.1	4.0	3.3	4.3	4.1	3.9
Fv/Fo, 7 day	4.3	4.3	4.1	3.7	4.0	3.2	4.2	3.9	4.3

## Data Availability

Data is contained within the article or SAppendix A.

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
