# Peer review of "Overwintering Camelina and Canola/Rapeseed Show Promise for Improving Integrated Weed Management Approaches in the Upper Midwestern U.S."

_plants, 2023, doi:10.3390/plants12061329_

Round 1

Reviewer 1 Report

The manuscript by Chao et al. investigates the traits of freezing tolerance and subsequent weed suppression in winter canola and camelina in two Upper Midwestern locations. The study is well conducted, well written and presents meaningful findings that I feel will be of significance to the scientific community and growers for cover crop improvement and application. I have the following questions/comments that can be clarified or addressed.

Line 63-64 It is commonly known that canola outperformed camelina in yield and established markets. However, the claim of better oil quality traits in canola is not well supported as it depends on which specific oil components for comparison.

Line 329-332 I assume there were 75 canola accessions (not plants) that overwintered in Fargo and 75 out of a larger number of overwintered accessions in Morris were randomly selected for genotyping. Did all those 75 accessions from Fargo survive in the winter of Morris? What is the genetic diversity of those overwintered accessions? A diversity evaluation, i.e. genetic structure analysis could be conducted to clarify that.

Author Response

Please see the attached file "Response to reviewer 1_Plants 2023".

Reviewer 2 Report

It is unclear why the air temperature was lower in Fargo, the soil temperature was lower in Morris (lines 139-143, figure 2). Probably should take into account the depth of the snow cover?

Author Response

Please see the attached file "Response to reviewer 2_Plants 2023".
